# Association of Chronic Otitis Media with Sjogren’s Syndrome: A Case-Control Study

**DOI:** 10.3390/jpm13060903

**Published:** 2023-05-27

**Authors:** Tzong-Hann Yang, Alison H. Chang, Yen-Fu Cheng, Chin-Shyan Chen, Herng-Ching Lin

**Affiliations:** 1Department of Otorhinolaryngology, Taipei City Hospital, Taipei 110, Taiwan; tzonghannyang@gmail.com; 2Department of Speech, Language and Audiology, National Taipei University of Nursing and Health, Taipei 112, Taiwan; 3Department of Otorhinolaryngology, School of Medicine, National Yang-Ming Chiao Tung University, Taipei 112, Taiwan; entist@gmail.com; 4Center of General Education, University of Taipei, Taipei 112, Taiwan; 5Research Center of Sleep Medicine, College of Medicine, Taipei Medical University, Taipei 110, Taiwan; stan@mail.ntpu.edu.tw; 6Department Physical Therapy and Human Movement Sciences, Feinberg School of Medicine, Northwestern University, Chicago, IL 60611, USA; hsini@northwestern.edu; 7Department of Medical Research, Taipei Veterans General Hospital, Taipei 112, Taiwan; 8Department of Otolaryngology-Head and Neck Surgery, Taipei Veterans General Hospital, Taipei 112, Taiwan; 9Institute of Brain Science, National Yang Ming Chiao Tung University, Taipei 112, Taiwan; 10Department of Economics, National Taipei University, New Taipei City 237, Taiwan; 11School of Health Care Administration, College of Management, Taipei Medical University, Taipei 110, Taiwan; 12Research Center of Sleep Medicine, Taipei Medical University Hospital, Taipei 110, Taiwan

**Keywords:** chronic otitis media, epidemiology, Sjögren’s syndrome, case-control study

## Abstract

There is a paucity of large-scale population-based study whether patients with Sjögren’s syndrome are at higher risk of chronic otitis media. This study aimed to investigate the association of chronic otitis media with Sjögren’s syndrome by utilizing the representative dataset of the Taiwanese population. We identified 9473 patients with chronic otitis media as cases. We used propensity score matching to select 28,419 controls. We used multiple logistic regression analysis to examine the association of chronic otitis media with prior Sjögren’s syndrome after adjusting for age, sex, monthly income category, geographic location and urbanization level of the patient’s residence, allergic rhinitis, chronic rhinosinusitis and tonsillitis and adenoiditis. Chi-square tests showed a statistically significant difference in Sjögren’s syndrome between patients with chronic otitis media and controls (4.89% vs. 2.93%, *p* < 0.001). In addition, we found patients with chronic otitis media were more likely to have Sjögren’s syndrome (OR = 1.698, 95% CI = 1.509~1.910) relative to controls after adjusting for age, income, geographic location, residential urbanization level, allergic rhinitis, chronic rhinosinusitis and tonsillitis and adenoiditis. We also found that of the male patients, patients with chronic otitis media had a greater tendency to Sjögren’s syndrome than controls (adjusted OR = 1.982, 95% CI = 1.584~2.481). Similarly, a statistically significant association between Sjögren’s syndrome and chronic otitis media remains in female sampled patients (adjusted OR = 1.604, 95% CI = 1.396~1.842). We found that patients with Sjögren’s syndrome were associated with the occurrence of chronic otitis media. It may guide physicians as they counsel patients with Sjögren’s syndrome on the possibility of chronic otitis media occurrence.

## 1. Introduction

Chronic otitis media is a persistent or recurring inflammatory condition affecting the middle ear and mastoid cavity, characterized by chronic infection with or without an intact tympanic membrane. This condition can lead to long-term or permanent changes in the middle ear and mastoid structures, including middle ear effusion, perforation, atelectasis, retraction, tympanosclerosis, and cholesteatoma [1]. Common symptoms associated with chronic otitis media include hearing loss, a feeling of fullness in the ear (aural fullness), ear pain (otalgia), ear discharge (otorrhea), and occasionally, episodes of vertigo. Numerous factors have been identified as potential risk factors for developing chronic otitis media, such as a history of acute otitis media, family history of chronic otitis media, allergies/atopy, upper respiratory tract infections (including symptoms like cough, runny nose, nasal congestion, sore throat, or adenoiditis/adenoid hypertrophy), snoring, cigarette smoking, living in larger families, higher crowding index, poor nutrition, and genetic factors. Global studies have shown that the incidence rate of chronic suppurative otitis media is highest in the first year of life (15.40 per thousand) and decreases to its lowest value after 65 years of age (2.51) [2]. In addition, chronic otitis media often results in some degree of hearing loss, typically conductive and temporary, caused by a ruptured tympanic membrane or changes in the ossicular chain due to fixation or erosion resulting from the chronic inflammatory process [3]. Comprehensive treatment options for chronic otitis media may involve the use of antibiotics or ear surgery, as this condition can give rise to serious or life-threatening complications such as mastoiditis, facial nerve paresis, labyrinthitis, petrositis [4], meningitis, thrombophlebitis, and brain abscess [5]. It affects a significant number of individuals, estimated to be between 65 and 330 million people worldwide, causing substantial hearing impairment in more than half of the affected population and resulting in 21,000 deaths annually [2]. Chronic otitis media can significantly impact a patient’s quality of life [6].

Sjögren’s syndrome, the second most common systemic autoimmune connective tissue disease after rheumatoid arthritis, is characterized by the lymphocytic-mediated destruction of exocrine glands, leading to dryness of the mouth and eyes. The prevalence of Sjögren’s syndrome is approximately 61 cases per 100,000 inhabitants, predominantly affecting women aged 40–60 years [6]. The disease can manifest as primary Sjögren’s syndrome (pSS) [7] or secondary when it co-occurs with other autoimmune systemic diseases such as rheumatoid arthritis, systemic lupus erythematosus, or systemic sclerosis [8]. Sjögren’s syndrome can also cause significant morbidity and mortality by affecting extra glandular organs such as the respiratory tract, muscles, nervous system, kidneys, and skin [9]. It also poses an increased risk of non-Hodgkin B-cell lymphoma. Despite extensive research, the exact underlying causes and pathomechanisms of Sjögren’s syndrome remain poorly understood. The pathogenesis of the disease may involve theories such as autoimmune epithelitis and neuroendocrine mechanisms [10]. The destruction of exocrine gland epithelium leads to abnormal responses of B cells and T cells to various autoantigens, including Ro/SSA and La/SSB.

Eustachian tube dysfunction plays a significant role in developing chronic otitis media. When inflammation affects the oropharyngeal mucosa due to conditions like respiratory tract infections or allergic rhinitis (seasonal or perennial), it can blockage or improper functioning of the Eustachian tube. This dysfunction leads to fluid accumulation in the middle ear, providing an environment for bacterial or viral growth, which can cause chronic otitis media. Case reports have indicated that benign lymphoid masses causing unilateral Eustachian tube obstruction can lead to severe otitis media in patients with Sjögren’s syndrome [11,12]. Clinical studies have also found conductive or mixed hearing loss cases among patients with Sjögren’s syndrome [13,14]. However, there is a lack of large-scale population-based studies examining whether patients with Sjögren’s syndrome have a higher risk of developing chronic otitis media. Consequently, we conducted the present study using a representative dataset of the Taiwanese population to investigate the association between chronic otitis media and Sjögren’s syndrome. Additionally, we analyzed this association in different sex groups.

## 2. Methods

### 2.1. Database

The sampled patients for this case-control study were retrieved from Taiwan’s Longitudinal Health Insurance Database 2010. Taiwan initiated the National Health Insurance program in 1995. According to the Taiwan National Health Insurance Act, every Taiwanese citizen with household registration must enroll in the National Health Insurance program regardless of age, gender, marital status or employment status. The single-payer National Health Insurance program provides comprehensive medical care to all Taiwanese citizens from birth. The Taiwan National Health Insurance program is characterized by a national global budget, payroll tax financing, equity in access to health services, provisions of comprehensive and generous population coverage, and very low copayment (less than US$2 for clinic visits). 

To protect patient privacy and the health care providers of patients, Taiwan’s Ministry of Health and Welfare has some restrictions on accessing the Longitudinal Health Insurance Database 2010. Only a researcher or a clinician from a university, research institute, or hospital is eligible to apply for the Longitudinal Health Insurance Database 2010. In addition, the Longitudinal Health Insurance Database 2010 must be used for research purposes only. All applications are to be reviewed by Taiwan’s Ministry of Health and Welfare to assure data confidentiality and the rationality of the use Plenty of researchers in Taiwan have gained access to the Longitudinal Health Insurance Database 2010 for epidemiological studies of healthcare utilization.

The study was approved by the institutional review board of Taipei Medical University (TMU-JIRB N202301005) and complied with the Declaration of Helsinki. However, because we used deidentified administrative data, informed consent was waived.

### 2.2. Identification of Cases and Controls

The current study was designed as a case-control study, a type of epidemiological observational study. This case-control study started with the identification of cases with chronic otitis media. We first retrieved 10,963 patients with a first-time diagnosis of chronic otitis media (ICD-9-CM codes 381.10, 381.19, 381.20, 381.29, 381.3, 381.4, 382.1, 382.2, 382.3 or ICD-10-CM code H65.2, H65.3, H65.4, H66.1, H66.2, H66.3) during an ambulatory care visit including clinics and outpatient departments of hospitals between 1 January 2016 and 30 December 2018. We then defined the first date of receiving the diagnosis of chronic otitis media as the index date for cases. We further eliminated 910 patients under 20 years of age to limit the study sample to the adult population. In addition, we refined the inclusion criteria to include only patients with at least two different medical claims showing a diagnosis of chronic otitis media by board-certified otolaryngologists because of potential concerns about the validity of diagnosis coding. Ultimately, 9473 patients aged ≥20 years old who had diagnoses of chronic otitis media were recruited as cases in this study.

The controls were retrieved from the Longitudinal Health Insurance Database 2010, and the remaining 1,872,500 enrollees were from the Registry for Beneficiaries. In addition to the Registry for beneficiaries, the Longitudinal Health Insurance Database 2010 included the Registries for contracted beds, contracted specialty services, contracted medical facilities, board-certified specialists, medical personnel, catastrophic illness patients, medical services, and drug prescriptions. All enrollees who had ever received a diagnosis of chronic otitis media since the initiation of the NHI program in 1995 and those aged less than 20 were excluded from the control group. Out of the remaining 1,530,035 enrollees, we carried out a propensity-scored matching to select controls, using the variables of sociodemographic characteristics including age, sex, monthly income, geographic location and urbanization level of the patient’s residence and medical comorbidities including allergic rhinitis (ICD-9-CM code 477 or ICD-10-CM code J30), chronic rhinosinusitis (ICD-9-CM code 473 or ICD-10-CM code J32) and tonsillitis and adenoiditis (ICD-9-CM code 474 or ICD-10-CM code J35). 

Propensity-score matching is a quasi-experimental method in which the researcher uses statistical techniques to reduce confounding effects in observational studies. Matching can help us ensure that any differences between the patients with chronic otitis media and the selected controls did not arise from the differences in the selected matching variables, including sociodemographic characteristics, allergic rhinitis, chronic rhinosinusitis and tonsillitis and adenoiditis. In this study, monthly income was grouped into three categories: NT$0~15,840, NT$15,841~25,000, ≥NT$25,001 (the average exchange rate in 2021 was US$1 ≈ NT$29). The variable of the geographic region of the patient’s residence included the Northern, Central, Southern, and Eastern parts of Taiwan. The urbanization level of the patient’s residence was grouped into five levels (1 most urbanized and five least urbanized). The logistic regression model is the most commonly used method for calculating propensity scores. The selected variables, including sociodemographic characteristics and allergic rhinitis, chronic rhinosinusitis, and tonsillitis and adenoiditis, were then entered into a multivariable logistic regression model as predictors to estimate the propensity score for patients with chronic otitis media and each remaining enrollee. Once the propensity scores for patients with chronic otitis media and each remaining enrollee are calculated, patients with chronic otitis media are matched with remaining enrollees with similar propensity scores. However, the exact score-matched controls may not be obtainable for every case, so patients with chronic otitis media are matched to the remaining enrollees with the closest propensity score within calipers. We set an apriori value for the caliper of 0.2. We used a controls-to-cases match ratio of 3:1 (*n* = 28,419). Controls were matched to a given patient with chronic otitis media if they had utilized any ambulatory care service in the index year of the chronic otitis media case. In addition, we defined the date of the control’s first visit to healthcare institutions, including clinics and hospitals, during the index year of their matched patients with chronic otitis media as their index date. As a result, this study consisted of 37,892 sampled patients, 9473 cases and 28,419 controls. 

### 2.3. Ascertainment of Exposure 

This study attempted to calculate the odds ratio of a prior diagnosis of Sjögren’s syndrome for patients with chronic otitis media compared to controls. The majority of case-control studies determine exposure from personal recall using self-report questionnaires. However, long-term recall might be less reliable. The present study identified Sjögren’s syndrome cases by ICD-9-CM code 710.2 or ICD-10-CM code M35.0 from medical claims of the Longitudinal Health Insurance Database 2010. Medical claims are usually more accurate and reliable relative to one that relies only on memory. Therefore, we included a patient with Sjögren’s syndrome only if they had ≥2 medical claims suggesting a diagnosis of Sjögren’s syndrome within the three years before their index date to assure the validity of the diagnoses of Sjögren’s syndrome.

### 2.4. Statistical Analysis

The SAS system (SAS for Windows, V, 9.4) for statistical analyses was used for descriptive analyses and inferential analyses in this study. We conducted χ^2^ and t tests to examine differences in the distributions of baseline information, including sociodemographic characteristics, allergic rhinitis, chronic rhinosinusitis, tonsillitis and adenoiditis between patients with chronic otitis media and controls. In addition, we performed multiple logistic regression analyses to investigate the association between chronic otitis media and prior Sjögren’s syndrome after taking into consideration sociodemographic characteristics, allergic rhinitis, chronic rhinosinusitis and tonsillitis and adenoiditis. We further analyzed the association between chronic otitis media and prior Sjögren’s syndrome according to sex group, adjusting for sociodemographic characteristics, allergic rhinitis, chronic rhinosinusitis, tonsillitis, and adenoiditis. This study employed *p* ≤ 0.05 to assess statistical significance.

## 3. Results

The distributions of sociodemographic characteristics and the prevalence of medical comorbidities of patients with chronic otitis media and controls are shown in Table 1. After performing propensity-score matching, we found that the mean age was 54.0 years old, with a standard deviation of 17.4 years for the sampled patients. Of the 37,892 sampled patients, 46.6% were males. In addition, the overwhelming majority of the sampled patients (53.1%) resided within communities in northern Taiwan, and less than 1.9% resided in southern Taiwan. The distribution of the level of urbanization revealed that almost 60% of the sampled patients inhabited the neighbourhoods characterized as residential urbanization levels 1 and 2. The mean age was 54.1 (±17.4) and 54.0 (±17.4) years for patients with chronic otitis media and controls, respectively (*p* value = 0.650). Furthermore, χ^2^ tests suggested that there were no statistically significant differences between patients with chronic otitis media and controls in sex (*p* value = 0.774), monthly income (*p* = 0.999), geographic location (*p* value = 0.146) and residential urbanization level (*p* value = 0.934).

Table 1 also presents the prevalence of allergic rhinitis, chronic rhinosinusitis, tonsillitis, and adenoiditis among patients with chronic otitis media and controls. χ^2^ tests revealed that there were no differences in the prevalence of allergic rhinitis (36.5% vs. 36.5%, *p* value > 0.999), chronic rhinosinusitis (11.4% vs. 11.4%, *p* value > 0.999) and tonsillitis and adenoiditis (1.0% vs. 1.0%, *p* value > 0.999) between patients with chronic otitis media and controls.

Table 2 shows the prevalence rates of Sjögren’s syndrome among patients with chronic otitis media and controls. The prevalence rate of Sjögren’s syndrome was 3.42% among the 37,892 sampled patients. χ^2^ tests revealed a statistically significant difference in prior Sjögren’s syndrome between patients with chronic otitis media and controls (4.89% vs. 2.93%, *p* value < 0.001). In addition, we analyzed the association between prior Sjögren’s syndrome and chronic otitis media according to sex group. We found that the statistically significant association between prior Sjögren’s syndrome and chronic otitis media remained in males (2.97% vs. 1.50%, *p* value < 0.001) or females (6.56% vs. 4.18%, *p* value < 0.001) sampled patients. We further analyzed the data and found that the mean number of days between the diagnosis of Sjögren’s syndrome and the occurrence of chronic otitis media was 669 days, with a standard deviation of 573 days.

Table 2 also reveals the results of the association between chronic otitis media and Sjögren’s syndrome by using multiple logistic regression analysis. We found that the crude odds ratio for Sjögren’s syndrome for patients with chronic otitis media compared to controls was 1.700 (95% confidence interval = 1.513–1.909). Furthermore, patients with chronic otitis media were more likely to have a prior diagnosis of Sjögren’s syndrome (odds ratio = 1.698, 95% confidence interval = 1.509~1.910) relative to controls after adjusting for sociodemographic characteristics, allergic rhinitis, chronic rhinosinusitis and tonsillitis and adenoiditis. We also found that of the male patients, patients with chronic otitis media had a greater tendency of a prior diagnosis of Sjögren’s syndrome than controls (crude odds ratio = 2.001, 95% confidence interval = 1.604~2.506). The odds ratio was 1.982 (95% confidence interval = 1.584~2.481) for a prior diagnosis of male patients with chronic otitis media relative to male controls after adjusting for sociodemographic characteristics, allergic rhinitis, chronic rhinosinusitis, and tonsillitis and adenoiditis. Similarly, a statistically significant association between prior Sjögren’s syndrome and chronic otitis media was still observed in female sampled patients (crude odds ratio = 1.608, 95% confidence interval = 1.403~1.844). Furthermore, after the data correction of sociodemographic characteristics, allergic rhinitis, chronic rhinosinusitis, and tonsillitis and adenoiditis were considered in the multiple logistic regression, we found that female patients with chronic otitis media were more likely to have a diagnosis of Sjögren’s syndrome than female controls (adjusted odds ratio = 1.604, 95% confidence interval = 1.396~1.842). 

## 4. Discussion

There is a lack of consensus among researchers on the relationship between Sjögren’s syndrome and chronic otitis media, and limited research has been conducted on this topic. The majority of previous literature is made up of case series or case reports. This is the first study to use a nationwide population-based case-control approach to explore the association of chronic otitis media in patients with Sjögren’s syndrome. The study found that Sjögren’s syndrome is a significant risk factor for chronic otitis media, with a 1.68-fold higher odds, after adjusting for sociodemographic characteristics (age, monthly income, sex, geographic location, and urbanization level of the patient’s residence), allergic rhinitis, chronic rhinosinusitis and tonsillitis and adenoiditis. 

Our studies indicated that chronic otitis media is an extra glandular manifestation of Sjögren’s syndrome. In line with our research, Doig et al. found four of twenty-two patients with primary Sjögren’s syndrome to have conductive hearing loss, with two having exudative otitis media and two having ear drum perforation. The exudative otitis media recovered by removing the crust in the nasopharynx [13]. In addition, cases of serous otitis media due to obstruction of the Eustachian tube by a lymphoid mass in the nasopharynx have been reported [11,12]. Intriguingly, a case with severe bilateral otitis media was reported to fit the criteria of both IgG4-related Mikulicz disease and Sjögren’s syndrome, underscoring the heterogeneous nature of the diagnosis of Sjögren’s syndrome [15]. 

Although some studies have explored the association of chronic otitis media with Sjögren’s syndrome, the exact mechanism by which Sjögren’s syndrome leads to chronic otitis media is not fully understood. It is believed that both mucosal and squamous chronic otitis media may be caused by the following two mechanisms. First, chronic otitis media could result from blockages in the Eustachian tube due to various factors such as inflammatory nodules, crusts, or copious secretions in the nasopharynx. Second, the infection may be exacerbated by impaired mucociliary clearance and decreased immune defences in the mucosa of the tympanic cavity, Eustachian tube, and nasopharynx. Dryness of the mucous membranes in the Eustachian tube and nasopharynx, subsequently causing crusts or copious secretions covering the Eustachian tube, have been suggested as potential causes of exudative otitis media or ear drum perforation in primary Sjögren’s syndrome [13]. The Eustachian tube, which connects the tympanic cavity to the nasopharynx, helps to equalize air pressure across the tympanic membrane and remove mucus from the tympanic cavity. Chronic dysfunction of the Eustachian tube can lead to various types of otitis media [16]. Sjögren’s syndrome is an autoimmune epithelitis [17] that affects exocrine glands and hinders the clearance of chronic inflammatory infiltrates [18]. Studies have shown that Sjögren’s syndrome is associated with an increased risk of inflammation and infection in the upper [19,20] and lower [21] respiratory tract. It is also possible that Sjögren’s syndrome affects the airway and the Eustachian tube epithelium. Decreased secretions in the Eustachian tube may impair mucociliary function and lead to the production of thicker, more viscous mucus with prolonged transport time [22,23].

Additionally, the lack of aqueous secretions that contain IgA in the mucosa of the tympanic cavity, Eustachian tube, and nasopharynx may weaken immune defences and increase the risk of infection [24,25]. Furthermore, chronic otitis media might result from a higher likelihood of upper respiratory infections in patients with Sjögren’s syndrome treated with various immunosuppressive drug therapies [26] or Cevimeline [27]. Additionally, Neuro-Sjögren with cranial nerve involvement [28,29] might affect the nerves that supply the muscles that open the Eustachian tube, resulting in dysfunction of the Eustachian tube and causing chronic otitis media. Further investigations should focus on exploring the functionality of the mucosa within the tympanic cavity, Eustachian tube, and nasopharynx among patients with Sjögren’s syndrome. This should include an examination of mucociliary clearance as it relates to removing viruses, bacteria, and mucus, as well as the IgA content in animal models or individuals with Sjögren’s syndrome. Additional studies are needed to determine whether muscles such as the tensor palatini muscle connected to the Eustachian tube are involved in patients with Sjögren’s syndrome.

A key strength of this study is utilising a large and real-world sample of patients in a nationwide population-based dataset. The large sample size enabled us to detect a real difference in the odds of prior Sjögren’s syndrome in patients with chronic otitis media. Furthermore, it matched controls with greater precision and power. In addition, we benefited from the strengths of the Taiwan healthcare system, which has been easily accessible and affordable for all residents since the introduction of national health insurance (NHI) in 1995. This low copayment system reduces the risk of selection bias based on socioeconomic status or residential location.

Additionally, we collected data on diagnoses of prior Sjögren’s syndrome and chronic otitis media from all sources, as the NHI claims da-ta cover all healthcare utilization for all Taiwanese residents (approx. 23 million). The comprehensive and affordable healthcare provided by the NHI program allows patients with both severe and minor medical conditions, such as Sjögren’s syndrome and chronic otitis media, to seek medical attention without delay. Therefore, socioeconomics does not impact the recognition of SS and chronic otitis media. Using NHI claims data also avoids potential recall bias often present in self-report survey data. Our study design, a case-control study with controls selected through propensity score matching, further strengthens the validity of our findings and allows for causal inferences between Sjögren’s syndrome and chronic otitis media while minimizing selection and misclassification bias.

While our study has several strengths, there are also some limitations. The claim data in the Longitudinal Health Insurance Database 2010, similar to other claims data, are subject to coding errors. We used recorded chronic otitis media codes based on the ICD-9-CM and ICD-10-CM codes, which may not be as accurate as diagnoses made through a standardized clinical examination, such as otoscopy and audiology tests. This lack of standardization could lead to case misclassification bias, such as mistaking acute otitis media for chronic otitis media. However, such misclassification bias is likely random, with similar proportions of patients with misdiagnosed otitis media among the patients with chronic otitis media and controls, and therefore unlikely to impact the validity of the findings.

Additionally, there may be limitations from inactive, healed, and active chronic otitis media. Patients with active chronic otitis media tend to seek medical help more than those with inactive or healed chronic otitis media. However, this selection bias is likely to be random, with similar proportions of patients with active chronic otitis media among the patients with chronic otitis media and controls, and therefore unlikely to bias the present findings. Another limitation is the lack of data on confounding variables such as family history, lifestyle, alcohol consumption, dietary factors, body mass index, occupation, environmental factors (for example, exposure to tobacco smoke [30] or housing [31]), genetic factors, race, manual workers [32], and laboratory data related to chronic otitis media and Sjögren’s syndrome. In addition, although our study is population-based, our findings may not be generalizable to other regions or countries due to differences in ethnicity and living environment. Almost all patients included in this case-control study were of Chinese ethnicity. Finally, one meta-analysis concluded that allergy, snoring, second-hand smoke, and low social status were risk factors for chronic otitis media [30]. Although the present study has taken sociodemographic characteristics, allergic rhinitis, chronic rhinosinusitis and tonsillitis and adenoiditis into consideration in the study design, the Longitudinal Health Insurance Database 2010 does not provide information on snoring and second-hand smoke. Further large-scale epidemiological studies are recommended to clarify the association between chronic otitis media and Sjögren’s syndrome by considering snore and second-hand smoke. 

## 5. Conclusions

In conclusion, this nationwide population-based case-control study found that Sjögren’s syndrome was associated with the occurrence of chronic otitis media; the odds ratio for Sjögren’s syndrome for patients with chronic otitis media compared to controls was 1.698. It may guide physicians as they counsel patients with Sjögren’s syndrome on the possibility of chronic otitis media occurrence. Further epidemiological studies are suggested to confirm the association of chronic otitis media with prior Sjögren’s syndrome in other regions or countries by considering environmental factors and family history.

## Figures and Tables

**Table 1 jpm-13-00903-t001:** Demographic characteristics of patients with chronic otitis media and controls in Taiwan (*n* = 37,892).

Variable	Patients with Chronic Otitis Media (*n* = 9473)		Controls (*n =* 28,419)		*p* Value
	Total No.	%	Total No.	%	
Males	4414	46.6	13,242	46.6	>0.999
Age, Mean (SD)	54.1 (17.4)		54.0 (17.4)		0.650
Monthly Income					0.774
<NT$1~15,841	1749	18.5	5154	18.1	
NT$15,841~25,000	3384	35.7	10,201	35.9	
≥NT$25,001	4340	45.8	13,064	46.0	
Geographic region					0.146
Northern	4971	52.5	15,150	53.3	
Central	2442	25.8	7079	24.9	
Southern	1880	19.9	5714	20.1	
Eastern	180	1.9	476	1.7	
Urbanization level					0.934
1 (most urbanized)	2830	29.9	8630	30.4	
2	2829	29.9	8424	29.6	
3	1610	17.0	4789	16.9	
4	1174	12.4	3496	12.3	
5 (least urbanized)	1030	10.9	3080	10.8	
Allergic rhinitis	3455	36.5	10,365	36.5	>0.999
Chronic rhinosinusitis	1084	11.4	3252	11.4	>0.999
Tonsillitis & Adenoiditis	90	1.0	270	1.0	>0.999

SD = standard deviation.

**Table 2 jpm-13-00903-t002:** Crude and covariate-adjusted odds ratios for prior Sjogren syndrome among patients with chronic otitis media case vs. controls by sex.

Variable	Total	
	Patients with Chronic Otitis Media Case (*n* = 9473)	Controls (*n =* 28,419)
Prior Sjogren syndrome	N, %	N, %
Yes	463 (4.89)	834 (2.93)
Crude OR (95% CI)	1.700 * (1.513~1.909)	1.000
Adjusted OR ^a^ (95% CI)	1.698 * (1.509~1.910)	1.000
	Males	
Yes	131 (2.97)	199 (1.5)
Crude OR (95% CI)	2.001 * (1.604~2.506)	1.000
Adjusted OR ^a^ (95% CI)	1.982 * (1.584~2.481)	1.000
	Females	
Yes	332 (6.56)	635 (4.18)
Crude OR (95% CI)	1.608 * (1.403~1.844)	1.000
Adjusted OR ^a^ (95% CI)	1.604 * (1.396~1.842)	1.000

Notes: CI = confidence interval; OR = odds ratio; ^a^ Adjustment for age, sex, monthly income, geographic location, urbanization level, allergic rhinitis, chronic rhinosinusitis and tonsillitis and adenoiditis; * *p* < 0.001.

## Data Availability

Data from the National Health Insurance Research Database, now managed by the Health and Welfare Data Science Center (HWDC), can be obtained by interested researchers through a formal application process addressed to the HWDC, Department of Statistics, Ministry of Health and Welfare, Taiwan (https://dep.mohw.gov.tw/DOS/lp-2506-113.html, accessed on 2 January 2022).

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
