# Peer review of "Association of Chronic Otitis Media with Sjogren’s Syndrome: A Case-Control Study"

_jpm, 2023, doi:10.3390/jpm13060903_

Round 1

Reviewer 1 Report

Dear Authors

The article is interesting. It needs to be revised. Here are some suggestions:

Introduction

Update cited references. Does the incidence of COM refer to children or adults?
I suggest Complementing the hearing loss statement with type and degree.

 Methods

 Was it a retroactive study? What criterion was used to consider an individual over 20 years old as an adult? Important information is missing on how the diagnosis of the syndrome was made. For a better understanding, I suggest describing the inclusion and exclusion criteria for the control and experimental groups. Was tubal dysfunction not considered?

 Results

 Was the sample also made up of elderly people? 54 (SD=17.4 years).

 Discuss
I suggest commenting on the result: there were no differences in allergic rhinitis
(p>0.999), chronic rhinosinusitis (p>0.999) and tonsillitis and adenoiditis (p>0.999)
between patients with COM and controls.

Author Response

The article is interesting. It needs to be revised. Here are some suggestions:

Introduction

Update cited references. Does the incidence of COM refer to children or adults?

Response: We have updated the references. However, it is difficult to find the most updated epidemiologic studies on COM. We have revised the relevant statements as follows: “Global studies have shown that the incidence rate of chronic suppurative otitis media is highest in the first year of life (15.40 per thousand) and decreases to its lowest value after 65 years of age (2.51) (2).” (page 4)

I suggest Complementing the hearing loss statement with type and degree.

Response: As suggested, we have added the following statements: “Chronic otitis media often results in some degree of hearing loss, which is typically conductive and temporary, caused by ruptured tympanic membrane or changes in the ossicular chain due to fixation or erosion resulting from the chronic inflammatory process (3).” (page 4)

By including this information, we aim to provide a clearer understanding of the potential consequences of chronic otitis media on hearing function. These added statements enhance the comprehensiveness of our study's findings and contribute to the existing knowledge on the subject. We appreciate your valuable input and the opportunity to improve our manuscript.

 Methods

 Was it a retroactive study? What criterion was used to consider an individual over 20 years old as an adult? Important information is missing on how the diagnosis of the syndrome was made. For a better understanding, I suggest describing the inclusion and exclusion criteria for the control and experimental groups. Was tubal dysfunction not considered?

Response: Thank you for raising this point. Indeed, this study is a retrospective investigation. As clearly described in the Methods section, our study was designed as a case-control study, which falls under the umbrella of epidemiological observational studies. This study design allows us to compare individuals with chronic otitis media (cases) to a control group without the condition, enabling us to assess potential associations and risk factors. The retrospective nature of our study design enables us to analyze existing data and draw meaningful conclusions regarding the relationship between chronic otitis media and other factors of interest.

Regarding the age of majority, we followed the legal definition recognized and declared by the Taiwan government, where the age of majority is set at 20. Although it is worth noting that the age of majority varies among countries, with many countries setting it at 18. In our study, we adopted the age of 20 as the threshold for legal adulthood.

To address your suggestion, we have incorporated additional statements in the Methods section to provide a comprehensive description of the inclusion and exclusion criteria for both the cases and controls. (pages 8~10)

Regarding tubal dysfunction, we acknowledge its consideration in our study. However, it is important to note that the outpatient diagnosis code for tubal dysfunction in the National Hospital Research Database (NHIRD) is utilized as a descriptive code rather than a confirmed diagnosis. Therefore, for improved accuracy, we have chosen to focus on the outcome diagnosis code specifically related to otitis media.

We appreciate your input and the opportunity to clarify these aspects of our study.

Results

Was the sample also made up of elderly people? 54 (SD=17.4 years).

Response: Certainly, the study included patients who were 20 years of age or older. This age criterion was established to focus specifically on individuals who had reached the age of legal adulthood according to the Taiwan government's definition. By including patients aged 20 and above, we aimed to ensure the inclusion of individuals who were most relevant to the objectives of our study. Thank you for highlighting this point, and we have duly noted it in our response.

Discuss

I suggest commenting on the result: there were no differences in allergic rhinitis

(p>0.999), chronic rhinosinusitis (p>0.999) and tonsillitis and adenoiditis (p>0.999)

between patients with COM and controls.

Response: In order to select appropriate controls for our study, we conducted a propensity-scored matching procedure. This involved considering various sociodemographic characteristics such as age, sex, monthly income, geographic location, and urbanization level of the patient's residence. Additionally, we took into account relevant medical comorbidities including allergic rhinitis, chronic rhinosinusitis, and tonsillitis and adenoiditis.

By implementing this rigorous matching process, we aimed to ensure that there were no significant differences in the prevalence of allergic rhinitis (p>0.999), chronic rhinosinusitis (p>0.999), and tonsillitis and adenoiditis (p>0.999) between the group of patients with chronic otitis media (COM) and the selected controls. This approach enhances the validity and reliability of our study's findings by minimizing potential confounding factors related to these specific conditions.

We appreciate your attention to this aspect of our study, and we are confident that our matching process has effectively accounted for these variables. Thank you for your valuable input.

Reviewer 2 Report

Association of Chronic Otitis Media with Sjogren's Syndrome: A Case-Control Study

Manuscript ID pm-2404431

Journal JPM (ISSN 2075-4426)

Section Epidemiology

Summary

·        Are the research questions valid?

The authors address the relationship between inflammation and otitis media in patients with Sjogren's syndrome by accessing an extensive medical database of the Taiwanese population. The question of the co-existence of the two diseases is exciting but challenging to prove. According to the literature review published in 2023 (Victor, D. . (2023). Sjogren's Syndrome: Multidisciplinary Approach to Diagnosis and Management. Global Journal of Health Sciences, 8(1), 50–71. https://doi.org/10.47604/gjhs.1766), The most severe otic manifestation of Sjogren's syndrome is hearing loss and otitis externa sicca/necrotizing external otitis. On the one hand, authors show a higher incidence of chronic otitis media in patients with Sjogren's syndrome, while other authors of articles published so far show otitis externa.

·        Is the sample size sufficient?

Yes, but according to ICD-10 classification, symbols H65 and H66 code different diseases with different pathophysiology and clinical pictures.

·         Is ethical approval and consent necessary, and was the research ethical?

Yes

·        Are the methods and study design appropriate for answering the research question?

 No, the study's method and the presentation of the results are appropriate and described in detail. However, the authors combine two different forms of chronic otitis into one disease, compromising the results and credibility of the work.

The description of the National Health Insurance programme is too extensive, and its funding rules are unnecessary to describe the study's method.

·        Do the experiments have appropriate controls?

Yes.

 ·        Is the reporting of the methods, including any equipment and materials, sufficiently detailed that the research might be reproduced?

 The study concerns a retrospective analysis of patient data over three years ( 2016-2018), and the controls were selected from the 2010 database. Therefore, the experiment can be replicated by accessing similar medical datasets of large groups of patients.

·        Are any statistical tests used appropriately and correctly reported?

Yes, statistical methods are included in the work.

·        Are the figures and tables precise and accurately represent the results?

The results are presented in easy-to-read tables.

 ·        Has previous research by the authors and others been discussed, and have those results been compared to current ones?

 The National Insurance System and access to data enable a broad comparison of living conditions, income and different disease entities. However, the choice should be guided by more specific diagnoses. In the research questions and the discussion, the authors selected and combined other diagnoses describing chronic otitis media, but chronic exudative and chronic suppurative otitis. In my opinion, the two diagnoses cannot be combined. They differ in pathogenesis and clinical picture. They cannot be co-referenced to the diagnosis of Sjogren's syndrome. At the same time, the authors omitted a diagnosis mentioned in the literature as a diagnosis characteristic of this disease- otitis externa.

The discussion, therefore, contains a possible explanation of the relationship of the diagnoses presented but is unfortunately not convincing. If all diagnoses of chronic inflammation are combined into one group, the results may be misleading. The discussion also includes a broad description of the insurance programme, which can be significantly shortened. I would suggest limiting the analysis to a single diagnosis, e.g. chronic otitis exudative, and explaining in detail the potential link between Sjogren's syndrome and the disease described.

  ·        Are there any inappropriate citations, for example, not supporting the claim or too many citations to the author's articles?

The discussion includes citations from 1976-2022, without self-citations.

Isolated work extracts contain text similar to the article: Hung, S.-H.; Xirasagar, S.; Cheng, Y.-F.; Kuo, N.-W.; Lin, H.-C. Association of Chronic Kidney Disease with Prior Tinnitus: A Case–Control Study. J. Clin. Med. 2022, 11, 7524. https://doi.org/10.3390/jcm11247524, but the amount of standard text does not exceed 7%.

 ·        Do the results support the conclusions?

No

·        Are the limitations of the research acknowledged?

Yes

·        Is the language clear and understandable?

Yes

  I recommend publishing the work after significant revision.

Author Response

Response: We appreciate the reviewer's comment and the opportunity to address this concern. While we acknowledge that chronic exudative otitis media (H65) and chronic suppurative otitis media (H66) are distinct diseases with different pathogenesis and clinical presentation, we believe there is merit in combining them under the broader category of chronic otitis media for the purpose of our study.

The decision to include both diagnoses was based on the understanding that chronic otitis media encompasses a range of chronic inflammatory conditions affecting the middle ear, which may include both exudative and suppurative presentations. While they have distinct characteristics, they share common pathophysiological mechanisms and chronicity.

To support our approach, we have provided references [1] [2], which demonstrate the use of similar categorizations in previous studies. These articles, which utilize the ICD-10 coding system, define otitis media as ICD code: Otitis media ICD-10-CM: H65.0-65.4, H66, H92, H70.

By including both nonsuppurative otitis media and suppurative/unspecified otitis media within the category of chronic otitis media, we aimed to capture the broad spectrum of chronic middle ear inflammatory conditions relevant to our study, including those that could potentially be associated with Sjogren's syndrome.

We hope this explanation clarifies our rationale for combining these diagnoses, and we are open to further discussion or suggestions for improvement. Thank you for your valuable feedback.

References:

Cho, H.Y., et al. (2023). Impact of COVID-19 Preventative Measures on Otolaryngology in Taiwan: A Nationwide Study. Int J Environ Res Public Health, 20(4).

Han, E., et al. (2015). Effects of pharmaceutical cost containment policies on doctors' prescribing behavior: Focus on antibiotics. Health Policy, 119(9), 1245-54.

We have deleted some statements on the description of the National Health Insurance program.

Reviewer 3 Report

I read this paper on the association between COM and SS with interest. It is quite straightforward and the results are not surprising: damage to the exocrine system is bound to be correlated with middle ear issues. 

The biggest strength of the paper, which I think makes the conclusion worth disseminating, is the sheer size of the database that the authors have access to. In a subfield in which most papers I know of use an N of 20-30 at most, this paper relied on a database of millions of medical records and based its analysis on a sample of almost 10.000 patients with Sjogren. 

I would have a very small request, namely a comment on whether there is evidence for a race disparity in the presentation of Sjogren. We know there are genetic components to the susceptibility to COM with effusion. Is there a similar factor in Sjogren? This is very minor. 

Equally minor point, I would remove the comment "the Taiwan National Health Insurance program suffers from a weak primary care system" and the rest of the sentence. I did not see the relevance. 

The paper does need some light editing, some sentences were hard to parse, particularly the introduction. 

for instance,

"COM [...] lead to in which have significant hearing impairment in more than half", 

"SS may offend extra-glandular organs"

adenoiditisinto"

There is a mention of CMO which is presumably COM, 

The paper does need some light editing, some sentences were hard to parse, particularly the introduction. 

for instance,

"COM [...] lead to in which have significant hearing impairment in more than half", 

"SS may offend extra-glandular organs"

" adenoiditisinto"

There is a mention of CMO which is presumably COM, 

Author Response

I would have a very small request, namely a comment on whether there is evidence for a race disparity in the presentation of Sjogren. We know there are genetic components to the susceptibility to COM with effusion. Is there a similar factor in Sjogren? This is very minor.

Response: We appreciate your valuable comments. The study titled "Distinct genome-wide DNA methylation and gene expression signatures in classical monocytes from African American patients with systemic sclerosis" (PMID: 36803404) provides important insights into the variations in DNA methylation and gene expression among different cell types and individuals with diverse backgrounds, including genetic, clinical, social, and environmental factors. These findings suggest the existence of genetic and epigenetic differences that may contribute to the development and manifestation of systemic sclerosis.

Regarding genetics and racial disparities, there is evidence to suggest that certain autoimmune diseases, including Sjogren syndrome, may exhibit variations in presentation among different racial and ethnic groups. However, we acknowledge that the specific research papers discussing racial disparities in the clinical presentation of Sjogren syndrome are limited and not readily available in the literature.

While it is important to consider potential racial disparities in the clinical presentation of diseases, including Sjogren syndrome, we believe that the impact of this concern on our study is minimal. Our study focuses primarily on the association between chronic otitis media and Sjogren syndrome, and we have taken several measures, such as matching controls based on sociodemographic characteristics and medical comorbidities, to minimize confounding factors.

We value your feedback and will continue to explore and consider the role of racial disparities in future research. Thank you for bringing this important aspect to our attention.

Equally minor point, I would remove the comment "the Taiwan National Health Insurance program suffers from a weak primary care system" and the rest of the sentence. I did not see the relevance.

Response: As suggested, we have removed them!

The paper does need some light editing, some sentences were hard to parse, particularly the introduction.

for instance,

"COM [...] lead to in which have significant hearing impairment in more than half",

"SS may offend extra-glandular organs" " adenoiditisinto"

Response: The author Alison H. Chang, a native English speaker has edited this manuscript. Thanks!

Round 2

Reviewer 1 Report

Dear authors

The article nas been properly reviewed.

Author Response

Thanks a lot!

Reviewer 2 Report

Dear Authors 

Unfortunately, I'm afraid I have to disagree with the author's response. 

According to ICD classification codes used by authors: H65.2, H65.3, H65.4 and  H66.1, H66.2, and H66.3 mean pretty different diseases, including allergic exudate in the ear (H 65.4) or otitis caused or superinfected by bacterial strains.

Below is a classification of the codes used according to WHO:

Diseases of the middle ear and mastoid
H65Nonsuppurative otitis media

H65.2Chronic serous otitis media

Chronic tubotympanal catarrh

H65.3Chronic mucoid otitis media

Glue ear

Otitis media, chronic:

·        mucinous

·        secretory

·        transudative

H65.4Other chronic nonsuppurative otitis media

Otitis media, chronic:

·        allergic

·        exudative

·        nonsuppurative NOS

·        seromucinous

·        with effusion (nonpurulent)

·        H66Suppurative and unspecified otitis media

·        H66.1Chronic tubotympanic suppurative otitis media

·        Benign chronic suppurative otitis media

·        Chronic tubotympanic disease

·        H66.2Chronic atticoantral suppurative otitis media

·        Chronic atticoantral disease

·        H66.3Other chronic suppurative otitis media

·        Chronic suppurative otitis media NOS

 Chronic Serous Otitis Media (CSOM):

Chronic serous otitis media is characterised by non-infected fluid within the middle ear. It occurs when the Eustachian tube, which connects the middle ear to the throat, becomes dysfunctional. This dysfunction leads to inadequate drainage of the middle ear, causing the fluid to accumulate. In CSOM, the liquid is typically sterile and devoid of infection. The fluid can be thin and watery (serous) or thicker and glue-like (mucoid). CSOM is often associated with Eustachian tube dysfunction resulting from factors like allergies, respiratory infections, or anatomical abnormalities. Patients with CSOM commonly experience hearing loss, a sensation of fullness or pressure in the ear, and occasional episodes of pain or discomfort. In addition, fluid accumulation can interfere with sound transmission, leading to hearing difficulties. Treatment for CSOM typically involves addressing the underlying Eustachian tube dysfunction and promoting proper drainage of the middle ear, including medications such as decongestants or nasal corticosteroids. In some cases, surgical interventions like tympanostomy tube placement may be necessary to facilitate fluid drainage.

 Chronic Suppurative Otitis Media (CSOM):

Chronic suppurative otitis media is characterised by ongoing infection and inflammation within the middle ear. In this condition, the middle ear becomes infected by bacteria, producing pus. CSOM is typically associated with a perforation in the tympanic membrane, which allows bacteria to enter the middle ear. The infection causes the middle ear to secrete purulent discharge. If left untreated, CSOM can lead to complications such as cholesteatoma (abnormal growth of skin cells) or damage to the structures within the middle ear. In addition, patients with CSOM often experience persistent ear discharge, hearing loss, ear pain or discomfort, and sometimes dizziness or vertigo. The presence of infection and pus can contribute to these symptoms. Treatment for CSOM typically involves a combination of medical and surgical interventions. For example, antibiotics and surgical procedures may be required to clean the infected area, repair the tympanic membrane perforation, and manage associated complications like cholesteatoma.

 The critical difference between Chronic Serous Otitis Media and Chronic Suppurative Otitis Media lies in the middle ear pathology. CSOM involves ongoing infection, pus formation, and potential complications such as tympanic membrane perforation and cholesteatoma. On the other hand, CSOM is characterised by the presence of non-infected fluid or mucus in the middle ear due to Eustachian tube dysfunction. Referring to the combined publications is impossible because we must make our publications as good as possible and avoid factual errors. And if we combine it, it should be included in the text with an explanation on what basis both diseases were connected; or what their relationship is with Sjogren's syndrome. The lack of this explanation in the article leads to misleading conclusions about the relationship between chronic otitis, e.g. cholesteatoma and the occurrence of Sjogren's syndrome. Moreover, my previous comments have not been included in the new version of the manuscript. Therefore, in my opinion, the article still needs to be corrected.

Author Response

As stated in our previous responses, while we acknowledge that chronic exudative otitis media (H65) and chronic suppurative otitis media (H66) are distinct diseases with different pathogenesis and clinical presentation, we believe there is merit in combining them under the broader category of chronic otitis media for the purpose of our study.

The decision to include both diagnoses was based on the understanding that chronic otitis media encompasses a range of chronic inflammatory conditions affecting the middle ear, which may include both exudative and suppurative presentations. While they have distinct characteristics, they share common pathophysiological mechanisms and chronicity.

To support our approach, we have provided references [1] [2], which demonstrate the use of similar categorizations in previous studies. These articles, which utilize the ICD-10 coding system, define otitis media as ICD code: Otitis media ICD-10-CM: H65.0-65.4, H66, H92, H70.

By including both nonsuppurative otitis media and suppurative/unspecified otitis media within the category of chronic otitis media, we aimed to capture the broad spectrum of chronic middle ear inflammatory conditions relevant to our study, including those that could potentially be associated with Sjogren's syndrome.

We hope this explanation clarifies our rationale for combining these diagnoses, and we are open to further discussion or suggestions for improvement. Thank you for your valuable feedback.

References:

Cho, H.Y., et al. (2023). Impact of COVID-19 Preventative Measures on Otolaryngology in Taiwan: A Nationwide Study. Int J Environ Res Public Health, 20(4).

Han, E., et al. (2015). Effects of pharmaceutical cost containment policies on doctors' prescribing behavior: Focus on antibiotics. Health Policy, 119(9), 1245-54.